# $A^2$: Efficient Automated Attacker for Boosting Adversarial Training

**Zhuoer Xu**[1,2], **Guanghui Zhu**[1]*, **Changhua Meng**[2], **Shiwen Cui**[2], **Zhenzhe Ying**[2],
**Weiqiang Wang**[2], **Ming Gu**[2], **Yihua Huang**[1]

[1]State Key Laboratory for Novel Software Technology, Nanjing University  [2]Tiansuan Lab, Ant Group
zhuoer.xu@smail.nju.edu.cn, zgh@nju.edu.cn,
{changhua.mch,donn.csw,zhenzhe.yzz,weiqiang.wwq,guming.mg}@antgroup.com,
yhuang@nju.edu.cn

## Abstract

Based on the significant improvement of model robustness by AT (Adversarial Training), various variants have been proposed to further boost the performance. Well-recognized methods have focused on different components of AT (e.g., designing loss functions and leveraging additional unlabeled data). It is generally accepted that stronger perturbations yield more robust models. However, how to generate stronger perturbations efficiently is still missed. In this paper, we propose an efficient automated attacker called $A^2$ to boost AT by generating the optimal perturbations on-the-fly during training. $A^2$ is a parameterized automated attacker to search in the attacker space for the best attacker against the defense model and examples. Extensive experiments across different datasets demonstrate that $A^2$ generates stronger perturbations with low extra cost and reliably improves the robustness of various AT methods against different attacks.

## 1 Introduction

DNNs (Deep Neural Networks) are extremely vulnerable to imperceptible perturbations despite their success in a wide variety of applications [He et al., 2016, Kenton and Toutanova, 2019, Guo et al., 2017]. In particular, adding small but carefully chosen deviations to the input, called adversarial perturbations, can cause DNNs to make incorrect predictions with high confidence. It indicates that models trained by minimizing the empirical risk are not intrinsically robust. To explicitly improve robustness, AT (Adversarial Training), where a defense model is trained on worst-case adversarial perturbations generated by an attacker, was developed and proved to be effective.

Based on the significant improvement of models' robustness, various methods have been proposed, which focus on different components of AT: analyzing the robustness of neural architectures Zagoruyko and Komodakis [2016], designing loss functions such as TRADES [Zhang et al., 2019] and MART [Wang et al., 2019], perturbing the model to regularize the loss landscape's flatness (i.e., AWP Wu et al. [2020]) and leveraging unlabeled data (i.e., RST [Carmon et al., 2019]). Gowal et al. [2020] compares the performance of each combination of most components and achieves the SOTA (state-of-the-art) performance.

All the above AT methods use PGD$^K$ [Madry et al., 2018], which is a $K$-step stack of FGSM [Goodfellow et al., 2015], as the attacker to generate perturbations for each example against the defense model. As the key of AT, stronger perturbations yield more robust models. However, there is a trade-off between the strength of the perturbation and the training efficiency. Increasing the attack step $K$ strengthens the perturbations, but linearly increases the training overhead [Gowal et al., 2020].

---

*Corresponding author.

Likewise, the huge overhead prevents the use of SOTA adversarial attackers [Croce and Hein, 2020, Yao et al., 2021]. To achieve a balance of robustness and efficiency, manually tuning the attacker (e.g., step size and attack method in each step) is of great concern. R+FGSM [Wong et al., 2019] first randomly initializes a small perturbation, and then applies FGSM with the tuned step size. Surprisingly, AT with R+FGSM is as effective as $\text{PGD}^K$ but has a significantly lower cost.

However, given the novel dataset, tuning the attacker manually is a challenging task requiring expert knowledge. Moreover, the best attacker is fine-grained to each example and current model during adversarial training. Manual coarse-grained tuning for the whole training (e.g., fixed attack method and step size for all examples) is sub-optimal and prevents further improvement of robustness.

Inspired by AutoML (Automated Machine Learning [Zoph and Le, 2017, Liu et al., 2018]), we propose an efficient automated attacker called $\text{A}^2$ to boost AT. $\text{A}^2$ is a parameterized attacker, which can automatically tune itself on-the-fly during training to generate worst-case perturbations for each example. First, we design a general attacker space by referring to existing attackers. The attacker space is stacked by one-step attacker cells. Each cell consists of a perturbation block and a step size block. Then, we employ a parameterized attacker to search for operations in each block and construct the attacker for each example that maximizes the model loss. Specifically, we leverage the attention mechanism to calculate the score of each operation. For continuous operations, we sum up the operations using the normalized scores as weights. For discrete operations, we use the reparameterization trick [Jang et al., 2017] to sample an operation from the corresponding block. In this way, the constructed attacker generates worst-case adversarial perturbations to train the defense model. Meanwhile, $\text{A}^2$ is differentiable and can be optimized with respect to the model loss by gradient descent.

We conduct extensive experiments to verify the effectiveness and efficiency of $\text{A}^2$. Compared with PGD, $\text{A}^2$ can find better perturbations for different models trained with various AT methods. The results demonstrate that 20-step $\text{A}^2$ generates better perturbations than $\text{PGD}^{100}$. Moreover, we combine $\text{A}^2$ with other AT variants, and the robustness of models with different architectures on various datasets is generally improved under strong attacks (e.g., classical C&W and SOTA AutoAttack). We also show that $\text{A}^2$ is insensitive to its hyperparameters.

To summarize, our main contributions can be highlighted as follows:

- We propose an efficient automated attacker called $\text{A}^2$, which can generate worst-case perturbations on-the-fly during training to improve robustness.
- In $\text{A}^2$, we design an attacker space by summarizing the existing attackers and employ a differentiable method to construct the most adversarial attacker for each example according to the attention mechanism.
- Extensive experimental results across different datasets and neural architectures demonstrate that $\text{A}^2$ improves the model's robustness by generating stronger perturbations in the *inner maximization*. Moreover, $\text{A}^2$ can be flexibly combined with different AT methods, showing good generality.

## 2 Preliminary: Adversarial Training

Let $D = (\mathbf{X}, Y) = \{(\mathbf{x}_i, y_i)\}_{i=1}^n$ be a dataset with $\mathbf{x}_i \in \mathbb{R}^d$ as a natural example and $y_i \in \{1, \dots, C\}$ as its associated label. We measure the performance of a DNN classifier $f$ parametrized with $\theta$ using a suitable loss function $l$, denoted as $\mathbb{E}_{(\mathbf{x}_i, y_i) \in D} [l(f_\theta(\mathbf{x}_i), y_i)]$. AT [Madry et al., 2018] formulates a saddle point problem whose goal is to find the model parameters $\theta$ that minimize the adversarial risk in the *outer minimization* (the example's index $i$ is omitted for brevity):

$$\underbrace{\min_\theta \mathbb{E}_{(\mathbf{x}, y) \in D} \left[ \overbrace{\max_{\delta \in \mathbb{S}} l\left(f_\theta\left(\mathbf{x} + \delta\right), y\right)}^{\text{inner maximization}} \right]}_{\text{outer minimization}} \tag{1}$$

where $\mathbb{S}$ defines the set of allowed perturbations. The perturbation is usually constrained by $L_p$ norm with a bound $\epsilon$, i.e. $\mathbb{S} = \{\delta | \|\delta\|_p \le \epsilon\}$.

The *inner maximization* aims to find an adversarial perturbation against the example that achieves a high loss for the defense model. However, it is NP-hard to find the optimum of the *inner maximization*. Various gradient-based attackers have been proposed to approximate its solution, and we classify them according to the number of steps in gradient ascent. One-step attackers [Goodfellow et al., 2015, Miyato et al., 2017] generate adversarial perturbations as:

$$\delta^* \approx \Pi_{\mathbb{S}} \, \eta \cdot \psi \left( \nabla_{\mathbf{x}} \right) \tag{2}$$

where $\nabla_{\mathbf{x}}$ is short for $\nabla_{\mathbf{x}} l \left( f_\theta \left( \mathbf{x} \right), y \right)$, $\eta$ is the step size, $\psi$ is a transformation function (e.g., *sgn* in FGSM and *Identity* in FGM) and $\Pi$ is the projection. Since such linearization attacks tend to be trapped in the non-smooth vicinity of the data point, R+FGSM initializes a small *random* perturbation to escape the vicinity and then applies FGSM. As a typical multi-step attacker, PGD$^K$ [Madry et al., 2018] can find better perturbations by $K$ step gradient ascent:

$$\mathbf{x}^{(k)} = \Pi_{\mathbf{x}+\mathbb{S}} \left( \mathbf{x}^{(k-1)} + \eta \cdot \psi \left( \nabla_{\mathbf{x}^{(k-1)}} \right) \right) \tag{3}$$

# 3 Methodology

## 3.1 Motivation

The key of AT is to generate perturbations in the *inner maximization*. Strong perturbation helps to improve robustness. It is generally accepted that the step $K$ used to solve the *inner maximization* correlates with the attacker's ability to generate stronger perturbation. However, larger $K$ leads to a linear increase in training overhead. Wong et al. [2019] suggests that with appropriate step size tuning and early stopping, one-step attackers yield models with the robustness that is comparable to much more expensive multi-step attackers. It indicates that hyperparameters, such as random initialization, step size, momentum, and early stopping, affect perturbation generation. From the perspective of effectiveness and efficiency, it is valuable to further improve robustness by tuning the attacker to strengthen perturbations.

However, manual tuning of perturbation generation for each example on-the-fly during training is impractical. To address this problem, we propose an *efficient automated attacker* to boost adversarial training by generating optimal perturbations on-the-fly during training.

## 3.2 Problem Formulation

Inspired by AutoML, we first design a general attacker space $\mathcal{A}$ by referring to existing attackers. Then, we employ an *automated attacker* parameterized by $\alpha$ to search in $\mathcal{A}$ and further construct an attacker against the example and the defense model $(\mathbf{x}, y, f_\theta)$. We abbreviate the perturbation generated by the constructed attacker as $\delta_\alpha$. Therefore, the goal of A$^2$ is to train a robust model using the perturbation generated by the constructed attackers through a bilevel optimization problem:

$$\min_\theta \mathbb{E}_{(\mathbf{x},y)\in D} \left[ l(f_\theta(\mathbf{x}+\delta_{\alpha^*}), y) \right]$$
$$\text{s.t. } \alpha^* = \arg\max_\alpha \mathbb{E}_{(\mathbf{x},y)\in D} \left[ l \left( f_\theta \left( \mathbf{x}+\delta_\alpha \right), y \right) \right] \tag{4}$$

On the attack side, we train $\alpha$ by SGD to make the defense model misclassify. On the defense side, we use $\alpha^*$ to construct the best attacker for each example and then generate perturbations to adversarially train the defense model.

## 3.3 Attacker Space

Revisiting most attackers, we find that the attacker can be viewed as a stack of one-step attackers consisting of an attack method and a step size. Thus, as shown in Figure 1(a), we design a general attacker space $\mathcal{A}$ consisting of $K$-step cells. The $k$-th cell is denoted as $C^{(k)}$, which is a one-step attacker consisting of the following two blocks:

**Perturbation Block $O_p$.** Typical attack methods (i.e., *FGM* and *FGSM*), attack methods with momentum (i.e., *FGMM* and *FGSMM*), random perturbations (i.e., *Gaussian* and *Uniform*), and the special *Identity* which enables the attacker to automatically early stop at a certain step like FAT [Zhang et al., 2020];

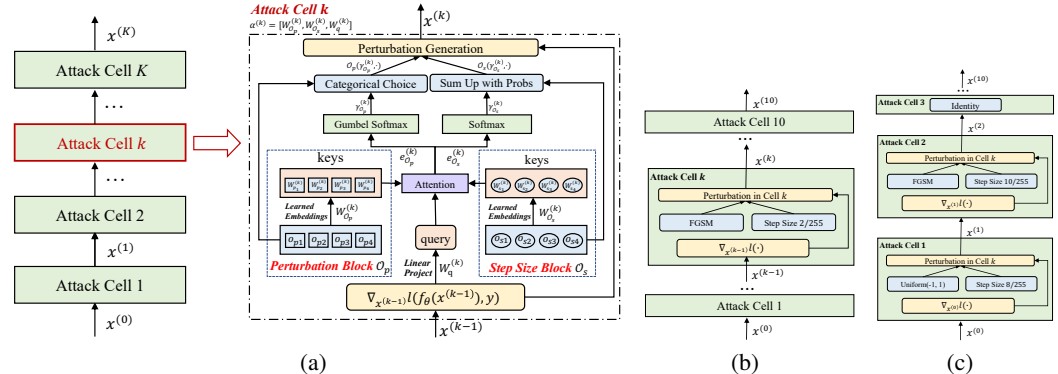

Figure 1: (a) Attacker Space of $A^2$; (b) $PGD^{10}$: *FGSM* and a fixed step size 2/255 in each cell; (c) R+FGSM: *Gaussian* in the first cell, *FGSM* in the second cell and *Identity* in the other cells.

**Step Size Block $O_s$.** $\{10^{-4} \cdot \eta, 10^{-3} \cdot \eta, 10^{-2} \cdot \eta, 10^{-1} \cdot \eta, \eta\}$, where $\eta$ is a hyperparameter related to the space of allowed perturbations $\mathbb{S}$.

Each block $O$ contains multiple operations. Let $o(\cdot)$ denote the operation, $\gamma^{(k)} = [\gamma_{O_p}^{(k)}, \gamma_{O_s}^{(k)}]$ denote the choice of operation in the $k$-th cell. The attack methods within $O_p^{(k)}$ are mutually exclusive. Thus, $\gamma_{O_p}^{(k)}$ is a one-hot vector. In contrast, the operations within $O_s^{(k)}$ are continuous. $\gamma_{O_s}^{(k)}$ is a normalized continuous vector, where each element represents the selection probability. To unify the categorical choice of attack methods and the probabilities over step sizes, the output of $O^{(k)}$ is expressed as a mixture based on $\gamma_O^{(k)}$:

$$\bar{O}(\gamma_O^{(k)}, \nabla_{\mathbf{x}^{(k-1)}}) = \sum_{o \in O^{(k)}} \gamma_o \cdot o(\nabla_{\mathbf{x}^{(k-1)}}) \tag{5}$$

where $\gamma_o$ denotes the weight of the operation $o$ in $\gamma_O^{(k)}$. Correspondingly, the one-step attacker of the $k$-th cell can be expressed as the joint of two blocks:

$$\bar{C}(\gamma^{(k)}, \nabla_{\mathbf{x}^{(k-1)}}) = \bar{O}_s(\gamma_{O_s}^{(k)}, \nabla_{\mathbf{x}^{(k-1)}}) \cdot \bar{O}_p(\gamma_{O_p}^{(k)}, \nabla_{\mathbf{x}^{(k-1)}}) \tag{6}$$

Moreover, the constructed attacker is a composition of attackers from each cell. In this way, we can cover common attackers in our space. For example, as shown in Figure 1(b), $PGD^K$ is obtained by selecting *FGSM* in each perturbation block. R+FGSM in Figure 1(c) is a case of selecting *Gaussian* in the first cell, *FGSM* in the second cell and *Identity* in the other cells.

**Analysis of $\mathcal{A}$** Considering there exist 7 attack methods in $O_p$ of each step, there are $7^K$ combinations of attack methods in the $K$-step attacker space. The exponential increasing combinations prevent the brute-force search. Moreover, the continuous step size $O_s$ is also part of the attacker space. Thus, we propose $A^2$ to search for the best attacker in $\mathcal{A}$ and generate adversarial perturbations efficiently.

### 3.4  Automated Attacker $A^2$

$A^2$ is used to construct the best attacker against $(\mathbf{x}, y, f_\theta)$ and its trainable parameters $\alpha$ include $W_{O_p}^{(k)}$, $W_{O_s}^{(k)}$, and $W_q^{(k)}$ where $k \in \{1, \dots, K\}$. As shown in Figure 1(a), we treat the current model and example as a query and the candidate operations as keys. Thus, the attention mechanism can be used to calculate the scores of operations within each block and the operations are selected based on their scores. Specifically, in the $k$-th cell, we take the gradient of the last step $\nabla_{\mathbf{x}^{(k-1)}} l\left(f_\theta\left(\mathbf{x}^{(k-1)}\right), y\right)$ as input and project it to a vector space as the query using $W_q^{(k)}$. Then, we use the trainable embedding table $W_O^{(k)}$ to convert the individual operations within $O^{(k)}$ to continuous keys. With the Scaled Dot-Product Attention [Vaswani et al., 2017], we compute the dot products of the query with each key as the score of the operation $o \in O$ in the $k$-th cell:

$$e_o^{(k)} = (\nabla_{\mathbf{x}^{(k-1)}} W_q^{(k)})^T W_o^{(k)} \tag{7}$$

**Perturbation Block.** The operations within $O_p^{(k)}$ are mutually exclusive. We sample an operation with the normalized scores as probabilities, i.e., $\gamma_{O_p}^{(k)} \sim softmax(e_{O_p}^{(k)})$.

**Step Size Block.** As the operations within $O_s^{(k)}$ are continuous values, we sum up the individual step sizes with the normalized scores as weights. For $o_s \in O_s^{(k)}$, the weight can be expressed as:

$$\gamma_{O_s}^{(k)} = \frac{\exp{(e_{o_s}^{(k)})}}{\sum_{o' \in O_s^{(k)}} \exp{(e_{o'}^{(k)})}} \tag{8}$$

### 3.5 Training of Automated Attacker

As mentioned above, we train $A^2$ to minimize the following objective by gradient descent:

$$\alpha^* = \arg\min_{\alpha} -\mathbb{E}_{(\mathbf{x},y) \in D} \left[ l\left(f_\theta\left(\mathbf{x} + \delta_\alpha\right), y\right)\right] \tag{9}$$

However, as a result of constructing the attacker by sampling in each perturbation block, the gradient of the loss w.r.t $\gamma_{O_p}$ is zero. To train $\alpha$, we use the reparameterization trick [Kingma and Welling, 2013] to transfer the randomness of sampling to the auxiliary noise and reformulate the objective function. For brevity, we omit the step index $k$.

Let $\gamma_{O_p} = \phi(\kappa, e_{O_p})$ be a differentiable transformation where $\kappa$ is an auxiliary noise variable with independent marginal $p(\kappa)$. In $A^2$, we sample noise from Gumbel Distribution, i.e., $\kappa \sim Gumbel(0)$ [Gumbel, 1954], and use Gumbel Softmax [Jang et al., 2017] as $\phi$ to smoothly approximate the expectation of loss [Maddison et al., 2014]. Specifically, $\phi(\kappa, e_{O_p}) = softmax\left((e_{O_p} + \kappa)/\tau\right)$ where $\tau$ is the temperature parameter. When $\tau \to 0$, the generated samples have the same distribution as $one\_hot(\arg\max_{o_p \in O_p}(e_{o_p} + \kappa_{o_p}))$.

Using the reparameterization trick, we can now form MC (Monte Carlo) estimates of the expectation of $A^2$'s loss $l$ for each example, which is differentiable, as follows:

$$
\begin{aligned}
&l\left(f_\theta(\mathbf{x} + \delta_\alpha), y\right) \\
&= \mathbb{E}_{\gamma_{O_p} \sim softmax(e_{O_p})} \left[ l\left(f_\theta(\mathbf{x} + \bar{C}([\gamma_{O_p}, \gamma_{O_s}], \nabla_\mathbf{x}), y\right)\right] \\
&= \mathbb{E}_{p(\kappa)} \left[ l\left(f_\theta(\mathbf{x} + \bar{C}([\phi(\kappa, e_{O_p}), \gamma_{O_s}], \nabla_\mathbf{x}), y\right)\right] \\
&\approx \frac{1}{M} \sum_{m=1}^{M} l\left(f_\theta\left(\mathbf{x} + \bar{C}([\phi(\kappa^{(m)}, e_{O_p}), \gamma_{O_s}], \nabla_\mathbf{x})\right), y\right)
\end{aligned} \tag{10}
$$

where $M$ is the number of samples. In practice, $M = 1$ can achieve good performance. In this way, we reformulate the MC approximation in Equation (10) of $l$ as the objective function $\hat{l}$.

Moreover, training $\alpha$ to convergence in each epoch can be prohibitive due to the expensive inner maximization in Equation (4). We use a simple approximation scheme following the common methods [Finn et al., 2017, Liu et al., 2018]:

$$\alpha^* \approx \alpha + \xi \nabla_\alpha \mathbb{E}_{(\mathbf{x},y) \in D} \left[ \hat{l}\left(f_\theta\left(\mathbf{x} + \delta_\alpha\right), y\right)\right] \tag{11}$$

where $\alpha$ denotes the current weights of the attacker and $\xi$ is the learning rate.

### 3.6 Framework of Adversarial Training with $A^2$

The overall procedure is shown in Algorithm 1. As in normal adversarial training, we generate perturbations in $K$ steps every batch and update the model parameters. The key difference is in Line 7. Benefiting from a parameterized automated attacker, we tune the discrete attack methods and continuous step sizes to generate adversarial perturbations. After optimizing the model parameters, we use Equation (11) to update $\alpha$ as an approximation to $\alpha^*$. Since $A^2$ focus on the *inner maximization*, it can be compatible with most adversarial training methods. For example, it is flexible to use the loss function of TRADES or MART for *outer minimization* in Line 10 (i.e., TRADES-$A^2$ and MART-$A^2$), or include early stopping in Line 5~9 as FAT.

---

**Algorithm 1** Adversarial Training with Automated Attacker (AT-A$^2$)

---

**Input**: Training examples $D$, perturbation bound $\epsilon$, the number of attack steps $K$

1: Initialize $\theta, \alpha$;
2: **for** epoch = $1, \ldots, N_{ep}$ **do**
3:      **for** minibatch $(\mathbf{X}, Y) \subset D$ **do**
4:         $\mathbf{X}^{(0)} \leftarrow \mathbf{X}$;
5:         **for** k = $1, \ldots, K$ **do**
6:            Calculate the gradient $\nabla_{\mathbf{X}^{(k-1)}}$;
7:            Construct $\delta_\alpha^{(k)} \in \mathbb{S}$ according to $\nabla_{\mathbf{X}^{(k-1)}}$ by $g_\alpha$;
8:            $\mathbf{X}^{(k)} = \mathbf{X}^{(k-1)} + \delta_\alpha^{(k)}$;
9:         **end for**
10:        Update $\theta$ with $\nabla_\theta \sum_{(\mathbf{x}, y)} l\left(f_\theta(\mathbf{x}^{(K)}), y\right)$;
11:        Update $\alpha$ by Equation (11);
12:      **end for**
13: **end for**

---

## 4 Experiments

We conduct extensive experiments on public datasets to answer the following questions: 1) Can A$^2$ generate stronger adversarial perturbations? 2) How effective is the adversarial training with A$^2$? 3) Is A$^2$ robust to hyperparameters? All experiments are run using GeForce RTX 3090 (GPU) and Intel(R) Xeon(R) Silver 4210 (CPU) instances.

### 4.1 Effectiveness of Automated Attacker (RQ1)

In this part, we fix the model $f_\theta$, train the automated attacker alone and investigate whether A$^2$ can generate more powerful perturbations compared to the commonly used PGD.

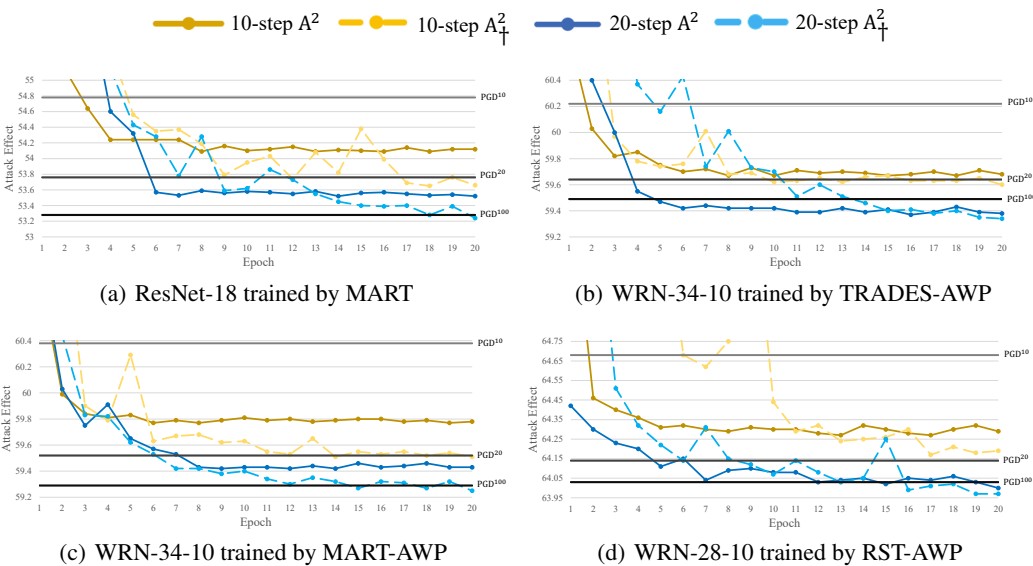

(a) ResNet-18 trained by MART

(b) WRN-34-10 trained by TRADES-AWP

(c) WRN-34-10 trained by MART-AWP

(d) WRN-28-10 trained by RST-AWP

Figure 2: Effect of adversarial perturbations generated by A$^2$ with the training epoch.

**Experimental Settings.** To demonstrate the generality, we choose different neural architectures (i.e., ResNet-18, WRN-34-10, and WRN-28-10) trained on CIFAR-10 by various AT methods: TRADES, MART, RST, and AWP. All trained models are open-source checkpoints. We choose PGD$^K$ with a random start $\delta^{(0)} \sim Uniform(-\epsilon, \epsilon)$ as baseline. All attacks are $L_\infty$-bounded with a total perturbation scale of $\epsilon = 8/255$. Since more attack steps generally improve the attack effect, we compare the automated attacker with PGD at different steps. Moreover, we try different step

Table 1: Comparison of attack effects (%, the lower the better) of multi-step PGD and $A^2$ in robust models. We run each method 5 times and show the average. The standard deviations are omitted as they are very small. The architecture of all defense models is WideResNet, except for MART whose architecture is ResNet-18.

| Defense | Natural | 10-step | | | 20-step | | | $PGD^{100}$ |
| --- | --- | --- | --- | --- | --- | --- | --- | --- |
| | | PGD | $A^2$ | $A^2_\dagger$ | PGD | $A^2$ | $A^2_\dagger$ | |
| MART[0] | 83.07 | 54.78 | 54.09 | 53.65 | 53.76 | 53.52 | **53.24** | 53.28 |
| TRADES-AWP[1] | 85.36 | 60.22 | 59.67 | 59.60 | 59.64 | 59.38 | **59.34** | 59.49 |
| MART-AWP[1] | 85.60 | 60.38 | 59.76 | 59.51 | 59.52 | 59.42 | **59.25** | 59.29 |
| RST-AWP[1] | 88.25 | 64.68 | 64.27 | 64.17 | 64.14 | 64.02 | **63.97** | 64.03 |

size blocks, i.e., $\eta$, in $A^2$: 1) $\eta = 2/255$, which is the same as the setting of step size in PGD; 2) $\eta = 8/255$, which is indicated by $A^2_\dagger$ and allows $A^2$ to search the whole $\epsilon$ bound each step. $A^2$ is trained using Adam [Kingma and Ba, 2015] with learning rate $10^{-3}$, weight decay $10^{-2}$ and other default hyperparameters.

**Attack Effect.** The training process of $A^2$ is shown in Figure 2. We take the first 20 epochs of the attack effects, compare them with PGD, and observe whether $A^2$ converges. In the early training stage, the random combinations of attack operations are much less effective. After 10~20 epochs, the effect of generated attacks is much more stable and effective. In practical automated adversarial training, the model and the automated attacker are trained iteratively. The fast convergence of $A^2$ ensures that the generated perturbations are strong enough. In addition, a larger $\eta$ achieves better attacks. As the steps increase, the effect diminishes and the training may fluctuate.

Table 1 reports the attack effects of PGD and $A^2$ with different steps. In the comparison of $A^2$ and PGD with different steps $K \in \{10, 20\}$, $A^2$ stably outperforms PGD. $A^2_\dagger$ constructs stronger attacks in the expanded search space. With the increase of steps, $A^2$ is more effective due to the combination of attack methods and the automated step size tuning. Due to the diminishing marginal effect, $PGD^{100}$ v.s. $PGD^{20}$ achieves less improvement than $PGD^{20}$ v.s. $PGD^{10}$. At 1/5 of the cost, the 20-step $A^2_\dagger$ finds better attacks using the optimized $\alpha$ than $PGD^{100}$. In summary, Table 1 verifies that $A^2$ stably outperforms PGD for the same step, and obtains better attacks compared to PGD, whose step size and attack method are fixed, with significantly lower cost.

**Overhead Analysis.** The overhead of $A^2$ is not significant compared to PGD. Both methods are close in terms of clock time. For WRN-34, PGD takes 19.75/147.09/287.76 seconds to generate 1/10/20 step attacks respectively. It demonstrates that more inner steps lead to a linear increase in time. Meanwhile, $A^2$ takes 157.61/302.51 seconds to generate the 10/20 step attack respectively. The main overhead remains in the forward computation and backward propagation of the defense model. Moreover, Section A.3 in Appendix shows that the total parameter size of $A^2$ is also acceptable.

## 4.2 Effectiveness of Adversarial Training with $A^2$ (RQ2)

In this part, we evaluate the robustness of our proposed AT-$A^2$ on different datasets against white-box and ensemble attacks. To verify that the stronger attacks generated by $A^2$ on-the-fly during training can improve robustness, we consider various adversarial training methods (i.e., AT, TRADES, MART, and AWP) without additional data across different datasets.

**Benchmark.** We conduct experiments on the baseline AT and the SOTA AWP with $A^2$ across three benchmark datasets to verify the generalization of $A^2$. We follow the settings in AWP: PreActResNet-18 trained for 200 epochs, $\epsilon = 8/255$ and $\gamma = 10^{-2}$ for AWP. The step size is 1/255 for SVHN and 2/255 for CIFAR-10 and CIFAR-100. For AT and AWP, the attacker used in training is $PGD^{10}$. The 10-step $A^2$ is trained with the same setting as in RQ1. $PGD^{20}$ is used for testing, and the test robustness is reported in Table 2. It shows that $A^2$, as a component focusing on the *inner maximization*,

---

[0]https://github.com/YisenWang/MART

[1]https://github.com/csdongxian/AWP

[2]https://github.com/zjfheart/Friendly-Adversarial-Training

Table 2: Test robustness (%, the higher the better) using PreActResNet-18 under $L_\infty$ threat model ("Best" means the highest robustness while "Last" means the robustness at the last epoch). Std. of 5 runs is omitted due to being small.

| Defense | SVHN | | CIFAR-10 | | CIFAR-100 | |
| --- | --- | --- | --- | --- | --- | --- |
| | Best | Last | Best | Last | Best | Last |
| AT | 53.36 | 44.49 | 52.79 | 44.44 | 27.22 | **20.82** |
| AT-A$^2$ | **56.76** | **44.75** | **52.96** | **44.59** | **28.14** | 20.28 |
| AWP | 59.12 | 55.87 | 55.39 | 54.73 | 30.71 | 30.28 |
| AWP-A$^2$ | **61.42** | **58.45** | **55.71** | **55.31** | **31.36** | **30.73** |

Table 3: Test robustness (%, the higher the better) on CIFAR-10 using WRN-34-10 under $L_\infty$ threat model ("Natural" denotes the accuracy on nature examples, and other columns indicate the accuracy on adversarial examples generated by different attacks). Std. of 5 runs is omitted due to being small.

| Defense | Natural | FGSM | PGD$^{20}$ | CW$_\infty$ | AutoAttack |
| --- | --- | --- | --- | --- | --- |
| AT | **87.30** | 56.10 | 52.68 | 50.73 | 47.04 |
| AT-A$^2$ | 84.54 | **63.72** | **54.68** | **51.17** | **48.36** |
| TRADES | 84.65 | 61.32 | 56.33 | 54.20 | 53.08 |
| TRADES-A$^2$ | **85.54** | **65.93** | **59.84** | **56.61** | **55.03** |
| MART | 84.17 | 61.61 | 57.88 | 54.58 | 51.10 |
| MART-A$^2$ | **84.53** | **63.73** | **59.57** | **54.66** | **52.38** |
| AWP | 85.57 | 62.90 | 58.14 | 55.96 | 54.04 |
| AWP-A$^2$ | **87.54** | **64.70** | **59.50** | **57.42** | **54.86** |

achieves better results on most datasets. Moreover, A$^2$ is generic and can boost the robustness of both baseline and SOTA AT methods.

**Robustness on WideResNet.** Furthermore, we train WRN-34-10 on CIFAR-10 with various AT methods (i.e., AT, TRADES, MART, and AWP) following their original papers and open-source codes[2]. All defense models are trained using SGD with momentum 0.9, weight decay $5 \times 10^{-4}$, and an initial learning rate of 0.1 that is divided by 10 at the 50%-th and 75%-th epoch. Except for 200 epochs in AWP, other AT methods train the model for 120 epochs. Simple data augmentations (i.e., 32x32 random crop with 4-pixel padding and random horizontal flip) are applied.

For white-box attack, we test FGSM, PGD$^{20}$ and CW$_\infty$ [Carlini and Wagner, 2017]. In addition, we test the robustness against the standard AutoAttack [Croce and Hein, 2020], which is a strong and reliable attacker to verify the robustness via an ensemble of diverse parameter-free attacks including three white-box attackers and a black-box attacker. Table 3 shows that A$^2$ reliably boosts AT variants against white-box and ensemble attacks. This verifies that A$^2$ is general for AT and improves adversarial robustness reliably rather than gradient obfuscation or masking.

Additionally, given the nature examples, AT performs better than AT-A$^2$. The main reason is that A$^2$ generates stronger perturbation for better robustness, which decreases the accuracy (i.e., 84.54). Many works (e.g., TRADES, MART, and AWP) use regularization to achieve the trade-off between robustness and accuracy. The regularization is also used to optimize the automated attacker. Thus, for other AT methods in Table 3, combining A$^2$ can achieve higher accuracy. Moreover, for WRN-34, the training time of AWP-A$^2$ is 970 s/epoch while the training time of AWP is 920 s/epoch. Thus, the additional overhead of A$^2$ is not significant.

### 4.3 Hyperparameters of A$^2$ (RQ3)

The hyperparameters of A$^2$ include the training hyperparameters and the design of the attacker space. The comparison of the attack effect with different hyperparameters is shown in Table 4. Overall, A$^2$ is robust to hyperparameters and performs better than PGD$^{10}$ and closely to PGD$^{20}$.

Table 4: Comparison of attack effects (%, the lower the better) of 10-step $A^2$ with different hyperparameters (The 10-step is omitted, $A^2_{p,q}$ is short for training the attacker using learning rate $\xi = 10^{-p}$ with the step size block $\eta = q/255$, ). Std. of 5 runs is omitted due to being small.

| Attack | TRADES-AWP[1] | MART-AWP[1] | RST-AWP[1] |
|---|---|---|---|
| $PGD^{10}$ | 60.22 | 60.38 | 64.68 |
| $PGD^{20}$ | 59.64 | 59.52 | 64.14 |
| $A^2_{3,2}$ | 59.67 | 59.76 | 64.27 |
| $A^2_{2,2}$ | 59.93 | 59.87 | 64.34 |
| $A^2_{4,2}$ | 59.76 | 59.78 | 64.29 |
| $A^2_{3,5}$ | 59.49 | 59.62 | 64.11 |
| $A^2_{3,8}$ | 59.53 | 59.53 | 64.17 |

**Training of $A^2$.** The effect of attacks with different learning rates $\xi$ is shown in the middle two rows of Table 4. Although Adam uses a dynamic learning rate, an excessive initial learning rate (i.e., $10^{-2}$) leads to sub-optimal.

**Attacker Space.** The influence of the attack step $K$ has been investigated in RQ1. As shown in the last two rows of Table 4, a larger step size $\eta$ increases the effectiveness of $A^2$. However, as shown in the training curve in Figure 2, larger $\eta$ introduces instability in the training of the attacker.

## 5 Related Work

### 5.1 Adversarial Learning

Many recent works [Goodfellow et al., 2015, Carlini and Wagner, 2017, Croce and Hein, 2020] have shown that DNNs are vulnerable to adversarial examples. Various defense strategies and models have been proposed to deal with the threat of adversarial examples. However, as proved in C&W [Carlini and Wagner, 2017], many works mistake gradient obfuscation or masking for adversarial robustness. AT [Madry et al., 2018] formulates a class of adversarial training methods for solving a saddle point problem (i.e., Equation (1)) and improves robustness reliably.

Based on AT, many works [Zhang et al., 2019, Wang et al., 2019, Wu et al., 2020] focusing on the components of *outer minimization* are introduced to further enhance performance. The *inner maximization* is also the goal of the adversarial attack, where $l$ is the 0-1 loss. Many works, e.g., FGSM [Goodfellow et al., 2015], C&W [Carlini and Wagner, 2017] and AutoAttack [Croce and Hein, 2020], have been proposed to attack DNNs and facilitate the development of adversarial training.

### 5.2 Automated Machine Learning

AutoML [Bergstra et al., 2011, Zoph and Le, 2017, Liu et al., 2018, Cubuk et al., 2019] aims to automate the parts of the machine learning pipeline that require expert solutions. For a particular domain, it is common practice to summarize a large search space of parameters and configurations based on expert experience and search for the optimal solutions using methods such as black-box optimization. During the search process, a certain metric is required to evaluate each solution. The same idea can be applied to adversarial learning. $A^3$ [Yao et al., 2021], which is also closely related to AutoML, automatically discovers an effective attacker on a given model.

## 6 Conclusion

In this work, we proposed $A^2$, to the best of our knowledge, the first adversarial training method which focuses on automated perturbation generation. In $A^2$, the attacker space is designed by summarizing the existing perturbations. Moreover, the parameterized automated attacker leverages the attention mechanism to choose the discrete attack method and the continuous step size and further generates adversarial perturbations. During training, the one-step approximation of the optimal automated attacker is used to generate the optimal perturbations on-the-fly for the model. The experimental

results show that $A^2$ generates stronger attacks with low extra cost and boosts the robustness of various AT methods reliably.

For future work, we plan to add the target loss of the *inner maximization* to the attacker space. We also plan to apply $A^2$ to enhance adversarial training for Natural Language Processing.

## Acknowledgments and Disclosure of Funding

This work was supported by the National Natural Science Foundation of China (#62102177 and #U1811461), the Natural Science Foundation of Jiangsu Province (#BK20210181), the Key Research and Development Program of Jiangsu Province (#BE2021729), Ant Group through Ant Research Program, and the Collaborative Innovation Center of Novel Software Technology and Industrialization, Jiangsu, China.

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
