# A Details in $A^2$

## A.1 Unify Magnitude of Perturbations

Perturbations generated by different operations in $O_p$ have different magnitudes and thus require different magnitudes of step size for different $o_p$. For example, *FGSM* generates perturbations with elements belonging to $\{-1, 0, 1\}$, while the perturbation generated by *FGM* is usually in the magnitude of $10^{-3}$. Obviously, they cannot use the same step size. To have a uniform effect of the step size block, we normalize the magnitude of other generated perturbations to be the same as *FGSM* (i.e., $\delta_{o_p} = \delta_{o_p} \cdot \frac{\|\delta_{FGSM}\|}{\|\delta_{o_p}\|}$). In this way, we find that other attack methods such as *FGM* can achieve good results with the same step size as *FGSM*.

## A.2 Temperature Parameter in *Softmax*

Since there is an order of magnitude difference in step size operations, the larger step size with the same score will dominate the output. For example, $0.7 \cdot 10^{-2}\eta + 0.3 \cdot \eta \approx 0.3\eta$. The output of the step size block is dominated by the operation $\eta$, despite the greater weight of $10^{-2} \cdot \eta$. To alleviate the problem, we use the temperature parameter $\tau$ in *softmax* to sharpen the distribution:

$$\gamma_{o_s}^{(k)} = \frac{\exp\left(e_{o_s}^{(k)}/\tau\right)}{\sum_{o' \in O_s} \exp\left(e_{o'}/\tau\right)} \tag{12}$$

where $o_s$ is an operation in $O_s$, and $e_{o_s}$ is its attention score. Through experiments, we set $\tau = 0.1$ to distinguish the preference for the step size in most cases.

## A.3 Overhead of $A^2$

Let the number of steps be $K$, the number of operations be $|O|$, the image size be $W \times H$ and the embedding size be $E$. The number of the attacker's parameter is $\mathbf{O}\left(K \cdot E \cdot (W \times H + |O|)\right)$. Specifically, the number of parameters for the attacker is 7873280, which is 17% of the model's parameters (i.e., 46160474). In each batch, there is only 1 forward calculation of all cells with 1 backpropagation. In comparison, the model requires $K$ forward calculations with backpropagation. Therefore, the additional computational overhead from the attacker is not significant in terms of the number of parameters and computations.

Moreover, PGD and $A^2$ are close in terms of clock time. For WRN-34, PGD takes 19.75/147.09/287.76 seconds to generate 1/10/20 step attacks respectively. It demonstrates that more inner steps lead to a linear increase in time. Meanwhile, $A^2$ takes 157.61/302.51 seconds to generate the 10/20 step attack respectively. The main overhead remains in the forward computation and backward propagation of the defense model. For WRN-34, the training time of AWP-$A^2$ is 970 s/epoch while the training time of AWP is 920 s/epoch.

In summary, the additional overhead of $A^2$ is not significant.

## A.4 Why No Mixture in $O_p$

Like most NAS methods in AutoML, the discrete selection in the perturbation block is more interpretable and robust (e.g., L1-Norm for feature selection and single path in NAS) than the mixture over possible solutions. Moreover, the mixture will incur more computational overhead and 7 times memory overhead due to 7 operations in $O_p$. Figure 3 shows an example of the generated attack on CIFAR-10, which can be migratable.

# B Addition Experiments

## B.1 Why use FGSM-based PGD in RQ1.

There are multiple single-step attack methods in $O_p$ for stacking as PGD, e.g., FGM-based PGD and FGSM-based PGD. The experimental results of the attack effect of PGD based on these attack methods demonstrate that FGSM-based PGD outperforms the stacking of other operations. Thus, we

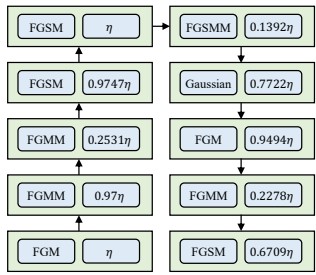

Figure 3: Example of generated attack on CIFAR-10.

choose FGSM-based PGD with a random start $\delta^{(0)} \sim Uniform(-\epsilon, \epsilon)$ as a baseline for comparison with the automated attacker.

## B.2  Number of samples $M$ in MC Approximation.

$M$ is an important hyperparameter that dictates the quality of MC approximation and the training overhead. We test the cases with $M \in \{1, 2, 5\}$ and achieve similar performance. Thus, we set $M$ to 1 and achieve good results with a significantly lower overhead.

## B.3  Generality of $A^2$ in White-Box Attacks

Table 5: Comparison of attack effects on CIFAR-10 (%, the lower the better) of PGD-based and $CW_\infty$-based attacks. The architecture of all defense models is WideResNet, except for MART whose architecture is ResNet-18.

|  | MART | TRADES-AWP | MART-AWP | RST-AWP |
|---|---|---|---|---|
| Natural | 83.07 | 85.36 | 85.60 | 88.25 |
| $PGD^{20}$ | 53.76 | 59.64 | 59.52 | 64.14 |
| $PGD^{20}$-$A^2$ | 53.24 | 59.34 | 59.25 | 63.97 |
| $CW_\infty$ | 49.97 | 57.07 | 56.44 | 61.82 |
| $CW_\infty$-$A^2$ | **49.82** | **56.98** | **55.81** | **61.30** |

In this part, we investigate whether $A^2$ is general to white-box attacks. As a more powerful attack method, $CW_\infty$-based attacks [Carlini and Wagner, 2017] stably outperform PGD-based attacks. For comparison with $CW_\infty$, we propose a variant of $A^2$ that uses $CW_\infty$ loss to generate perturbations and denote it as $CW_\infty$-$A^2$. The results in Table 5 show that $A^2$ is general and can improve the attack effect of PGD and $CW_\infty$ by combining attack methods and tuning the step size. Moreover, the additional overhead of $A^2$ is 5% to 10%, which is a rather acceptable trade-off.

## B.4  Robustness Against Transferable Black-Box Attacks

We investigate the robustness of $A^2$ against transferable black-box attacks. Table 6 provides test robustness on CIFAR-10 using ResNet-18. We adopt three transferable black-box attack methods: MI (momentum = 1) [Dong et al., 2018], DI [Xie et al., 2019], and TI [Dong et al., 2019]. The transferable attacks are generated by an ensemble of the above methods on three surrogate pre-trained models [2]: IncV3 (InceptionV3), VGG19, and DN201 (DenseNet201). Table 6 shows that AT boosts the robustness against transferable black-box attacks, and $A^2$ can further improve the adversarial robustness.

Table 6: Test robustness (%, the higher the better) on CIFAR-10 using ResNet-18 against transferable black-box attacks.

| | MI+DI+TI | | | |
| --- | --- | --- | --- | --- |
| | IncV3 | VGG19 | DN201 | PGD$^{20}$ |
| ResNet-18 | 16.12 | 7.37 | 5.35 | 0.02 |
| ResNet-18-AT | 61.98 | 60.81 | 59.63 | 52.79 |
| ResNet-18-AT-A$^2$ | **62.79** | **61.85** | **60.28** | **52.96** |

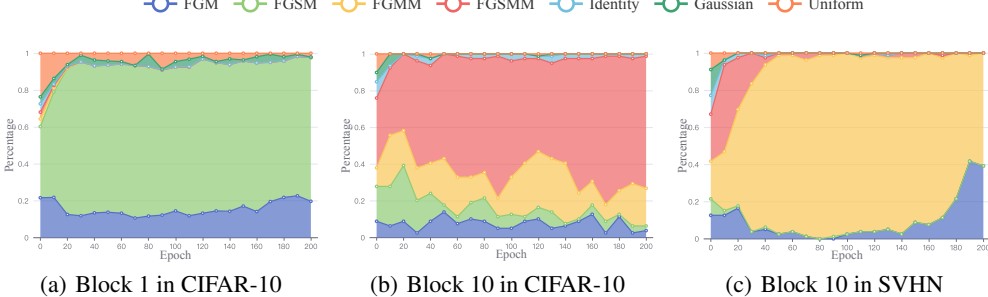

(a) Block 1 in CIFAR-10    (b) Block 10 in CIFAR-10    (c) Block 10 in SVHN

Figure 4: Distribution of attacks selected by perturbation blocks of A$^2$.

## B.5  A Closer Look at Selected Attacks

We analyze the selected attacks from the perspective of perturbation blocks with different steps and datasets.

The first and final perturbation blocks of 10-step A$^2$ in CIFAR-10 are chosen for analysis. Figure 4 shows the distribution of selected attacks of different perturbation blocks.

- **Perturbation Block 1:** A$^2$ tends to choose *FGM*, *FGSM*, and partially random methods as initialization in the first step. The momentum-based attack methods are quickly discarded as the gradient of the previous step is absent. *FGSM* is chosen more frequently due to its stronger attack on both foreground and background.

- **Perturbation Block 10:** The optimization of the victim model leads to changes in the distribution of selected attacks in the last block. In the early stage of training, the victim model is vulnerable. A$^2$ retains the diversity and plays the role of friendly attackers like FAT [Zhang et al., 2020]. At the end of training, A$^2$ prefers the momentum-based attacks (i.e., *FGSMM* and *FGMM*).

From the perspective of datasets, SVHN and CIFAR-10 prefer different attack methods. As shown in Figure 4(c), SVHN discards *FGSMM*, which is most frequently used in CIFAR-10, and pays more attention to *FGMM*. Moreover, SVHN rarely uses *Identity* compared with CIFAR-10 as its higher robustness accuracy requires more powerful perturbations.

In summary, A$^2$'s preference for selecting attacks in blocks varies according to the block step, dataset, and victim model.

---

[2]https://github.com/huyvnphan/PyTorch_CIFAR10