# OpenReview forum: "A2: Efficient Automated Attacker for Boosting Adversarial Training"
_NeurIPS.cc/2022/Conference — NeurIPS 2022 Accept_

### Official Review · Reviewer_rb9H · 2022-07-10

**Rating:** 7
**Confidence:** 4
**Soundness:** 4 excellent
**Presentation:** 3 good
**Contribution:** 4 excellent

**Summary:**

This paper presents a more efficient way for adversarial training – deciding attack algorithm parameter based on the idea of AutoML. In this framework, attack steps are split into step-wise cells, and the proposed method aims to predict the best parameter for each cell using attention mechanism. By fixing the parameters of each cell, the proposed framework degenrates into existing adversarial training method. The proposed method has been evaluated on three commonly used datasets: SVHN, CIFAR-10, CIFAR-100. Comprehensive experimental results demonstrate the effectiveness of the proposed method.

**Questions:**

My questions are listed in the weaknesses above. Generally this is a good paper and I did not find many questions.

**Limitations:**

By searching the keyword "limitation" I found no result in the submission, but I consider this a minor issue.
This paper focuses on defense side, and hence negative impact is not of concern.

**Strengths And Weaknesses:**

# Strengths

1. Traditional defense methods leverages multi-step PGD for creating adversarial examples to adversarially train a model. Such method is promising but poses very significant computational overhead. I've also gone through the pain of slow adversarial training. Hence, the motivation to create adversarial examples more effectively for training is valid and clear.

2. The proposed method is novel – introducing ideas of automl into the manual parameter tuning of attack algorithms for better results. Meanwhile, the method is clearly described and is not difficult to understand.

3. The proposed method is effective and clearly improves robust accuracy. Nowdays it's very hard to improve adversarial robustness on these standard benchmarks. Compared to the recent related works, the robustness improvement introduced by this method (such as Table 2) is already non-trivial.

4. The proposed method is compatible with other state-of-the-art defense methods like AT, TRADES, and AWP.

# Weaknesses

1. [important] Is the resulting adversarial perturbation clipped into the perturbation bound epsilon? The perturbation bound epsilont is reuiqred by Algorithm 1, but not explicitly used. Please clarify this and make it less confusing.

2. [minor suggestion.] It would be better to investigate wether the proposed method is still effective on larger scale datasets like ImageNet.

---

> ### Author Response · Authors · 2022-08-02
> **Response to Reviewer rb9H**
>
> Thank you very much for the positive feedback and the useful comments.
> We have addressed the issues in the revised version. The detailed response to each comment is as follows:
>
>
> > Weakness1: [important]Is the resulting adversarial perturbation clipped into the perturbation bound epsilon? The perturbation bound epsilon is required by Algorithm 1, but not explicitly used. Please clarify this and make it less confusing.
>
> Yes, as shown in `SupplementaryMaterial/auto_adv/cell.py/Line77`, all perturbations have been clipped into $\epsilon$-bound via `torch.clamp`.
> Thank you for pointing out this issue.
> In the revised version, we have highlighted this in `Algorithm1/Line7`.
>
>
> ---
>
> > Weakness2: [minor suggestion.] It would be better to investigate whether the proposed method is still effective on larger-scale datasets like ImageNet.
>
> Thank you for your suggestion.
> Currently, we are unable to obtain results due to the expensive running time.
> This will serve as our future work.

---

> > ### Comment · Reviewer_rb9H · 2022-08-06
> > **Response to Authors**
> >
> > I have read all reviewer comments as well as the author response. And I did not find a reason to reduce my original recommendation (7:accept). I agree that improving even 1% of adversarial robustness is already difficult enough at the current stage. It is highly suggested to release code for reproducing this work if it will be accepted.

---

> > > ### Author Response · Authors · 2022-08-07
> > > **Thank you for the acknowledgment**
> > >
> > > We sincerely appreciate your time and efforts in reviewing our paper.
> > > We truly thank you for the useful suggestions and the acknowledgment of $A^2$''s improvement.
> > > For reproducibility, we will release the code on Github if it is accepted.

---

### Official Review · Reviewer_WxDd · 2022-07-11

**Rating:** 6
**Confidence:** 4
**Soundness:** 4 excellent
**Presentation:** 2 fair
**Contribution:** 3 good

**Summary:**

This paper proposes a novel attack method that efficiently generates strong adversarial perturbations. Specifically, based on the idea of AutoML, the authors design an automated attacker $A^2$ that finds the best perturbation for each iteration. The main idea is to use an attention mechanism to score possible attacks in the attacker space, then sample the attack to perform based on the assigned scores. The authors challenge the problem of training attention mechanism via reparameterization trick. By training both the model parameter and the automated attacker, the authors propose an improved adversarial training method using $A^2$.

From a set of experiments, the paper answers three questions about $A^2$: the power of the attack method, its effect in adversarial training, and the robustness of $A^2$ training. The experimental results demonstrate that $A^2$ can increase the attack power and improve the adversarial training performance without too much overhead.


**Questions:**

1. This paper specifically uses $\nabla_{\mathbf{x}^{(k-1)}} l(f_\theta(\mathbf{x}^{(k-1)}, y)$ to produce a query. Is it a necessary choice, or is your design flexible enough to take other values as input? To provide more context, I’m considering a gradient-free version of $A^2$ and expect the same design to work with a different attacker space and a different type of query.

**Limitations:**

The authors discussed limitations and potential negative societal impact in the paper appropriately.

**Strengths And Weaknesses:**

Originality: To the best of my knowledge, the paper contains novel ideas.

Quality:

[[Strength]]
1. The construction looks sound, and the reparametrization trick seems reasonable to handle similar situations in model training.
2. The authors used various adversarial training methods, datasets, and attack methods for the adversarial training experiments, showing the proposed method's effectiveness.

[[Weakness]]
1. The authors can improve the experiment on the attack effectiveness by more comparisons to other attacks (other than PGD) such as CW and AutoAttack.

Clarity: The paper writing is clear, and there is no issue with the clarity of the paper.

Significance:

[[Strength]]
1. The result on the attack effectiveness shows that we can save the number of iterations by adapting $A^2$ in adversarial example generation.
2. The experiments contain a practical overhead analysis for people adapting $A^2$ in practice. Considering the additional power of the attack, I believe the overhead is reasonably small.

---

> ### Author Response · Authors · 2022-08-02
> **Response to Reviewer WxDd**
>
> We sincerely thank the positive feedback and useful suggestions from the reviewer. The revised version has been uploaded. Here is our response.
>
>
>
> > Question1: This paper specifically uses $\nabla_{x^{(k-1)}} l$ to produce a query. Is it a necessary choice, or is your design flexible enough to take other values as input? To provide more context, I’m considering a gradient-free version of $A^2$ and expect the same design to work with a different attacker space and a different type of query.
>
> $A^2$ is flexible to take other values as input, as long as the input can be extracted as queries and the victim model back-propagates the loss in a white-box setting.
>
> In fact, $\nabla_{x^{(k-1)}}$ is an unnecessary but good choice during adversarial training, which contains the information of the model and sample.
> We design such a query with reference to observable states and reward mechanisms in reinforcement learning.
> The more information the state (i.e., $\nabla_{x^{(k-1)}}$) and reward (i.e., backpropagation of loss) provide, the more efficient the automated attacker is.
>
> Furthermore, other black-box optimization algorithms need to be considered (e.g., Bayesian Optimization) in black-box settings where model gradients are not available.
>
>
> ---
>
> > Weakness1: The authors can improve the experiment on the attack effectiveness by more comparisons to other attacks (other than PGD) such as CW and AutoAttack.
>
>
> Many thanks for the suggestion.
>
> According to the suggestion, we have added the attack effect of CW in Appendix B.3 and investigate whether $A^2$ is general to other white-box attacks.
> As a more powerful attack method, $CW_{\infty}$-based attacks stably outperform PGD-based attacks.
> For comparison with $CW_{\infty}$, we propose a variant of $A^2$ that uses $CW_{\infty}$ loss $\nabla_{x^{(k)}} l_{cw}$ as input to generate perturbations and denote it as $CW_{\infty}$-$A^2$.
> The results show that $A^2$ is general and can improve the attack effect of PGD and $CW_{\infty}$ by combining attack methods and tuning the step size.
>
>
> | | MART | TRADES-AWP | MART-AWP | RST-AWP|
> |--|-----|-------|-----|----|
> |Natural|83.07|85.36|85.60|88.25|
> |$PGD^{20}$|53.76|59.64|59.52|64.14|
> |$PGD^{20}-A^{2}$ | 53.24 | 59.34 | 59.25 | 63.97|
> |$CW_{\infty}$ | 49.97 | 57.07 | 56.44 | 61.82|
> |$CW_{\infty}-A^{2}$ | **49.82** | **56.98** | **55.81** | **61.30** |

---

> > ### Comment · Reviewer_WxDd · 2022-08-06
> > **Respond to the authors**
> >
> > Thanks for the additional experiments on the CW attack effect. I also appreciate the authors' insight about relating the choice of $\nabla_{\mathbf{x}^{(k-1)}} l(f_\theta(\mathbf{x}^{(k-1)}, y)$ to the reinforcement learning context. However, I'm not very convinced about this paper's "high impact" (compared to the existing cutting-edge methods, the improvement seems to be subtle), so I decided to stay at my initial evaluation of 6.

---

> > > ### Author Response · Authors · 2022-08-06
> > > **Thank you for the acknowledgment**
> > >
> > > We sincerely appreciate your constructive comments. The comparison with CW and the design of the gradient-free version inspires us to explore $A^2$ in a more diverse attack space (e.g., CW and transfer-based black-box attack). As a plug-in, $A^2$ is used to boost various adversarial attack&training methods under the consideration of efficiency.

---

### Official Review · Reviewer_SX8F · 2022-07-11

**Rating:** 6
**Confidence:** 2
**Soundness:** 3 good
**Presentation:** 2 fair
**Contribution:** 4 excellent

**Summary:**

- The authors propose a new approach for the adversarial training of deep neural networks using their $A^2$ automated attacker. By using an efficient and stronger attack algorithm, the adversarial perturbations can result in more robust models trained using adversarial training. Their inner attack algorithm is inspired by AutoML and can select the attack parameters that generate the worst-case perturbations. The authors’ attack incurs a small increase in runtime compared to PGD.

**Questions:**

- Are the results from tables 2, 3, and 4 also run 5 times and averaged as well? Can we be certain that the $A^2$ approach shows a consistent improvement over normal adversarial training (and were not cherry-picked results)?
- Did the authors investigate the efficacy of their $A^2$ adversarial training approach against transferable black box attacks?
- I’m curious how frequently each attacker parameter is selected. Is there a combination of attack types or step sizes which are clearly selected a vast majority of the time? If so, does $A^2$ provide a considerable improvement over just using this combination of attack parameters? Or are there classes or datasets that are more vulnerable to certain attack combinations?


**Limitations:**

- The main limitations of this work can be addressed with the following suggestions:
	- I would suggest the authors log the distribution of selected attack parameters for each dataset and class.
	- The authors should ensure that the the results presented in tables 2-5 are reproducible. Though, it is understandable these experiments were run only once, given the expensive runtime of adversarial training.
	- Emphasizing or explaining the significance of the relatively small performance improvements afforded by $A^2$


**Strengths And Weaknesses:**

Strengths

- A new attack type that can be used to improve the robustness of models defended using adversarial training. The attack increases runtime by 5-7%, a rather acceptable trade-off.
- The results and experiments show a consistent improvement over previous work in both attacking models and using $A^2$ for adversarial training.
- Included analyses and explanations of certain results (e.g., natural vs. adversarial accuracy trade-off).

Weaknesses

- Most results only indicate small improvements of less than 2 percentage points.
- It is unclear whether results from tables 2, 3, and 4 are also run and averaged 5 times.
- Various minor typos and grammar issues: is developed -> was developed, PDG -> PGD, such linearization attack -> attacks tend, becompatible -> be compatible, etc. I would recommend the authors perform a thorough proofreading check.
- No investigation of whether different datasets, classes, or samples actually require a significant amount of fine-tuning to find optimal attack parameters.

---

> ### Author Response · Authors · 2022-08-02
> **Response to Reviewer SX8F**
>
> We thank the reviewer for the valuable comments. The revised version has been uploaded. We have added experiments on transferable black-box attacks and a discussion of selected attacks in Appendix. Moreover, We have fixed the grammatical errors in the revised version. Next, we introduce the responses point by point.
>
> ---
>
> > Q2: Did the authors investigate the efficacy of their adversarial training approach $A^2$ against transferable black box attacks?
>
> According to the suggestion, we further investigate the effectiveness of $A^2$ against transferable black-box attacks in Appendix B.4.
> The transferable attacks are generated by **an ensemble of three transferable black-box attack methods (MI with momentum = 1[1], DI[2], and TI[3]) with 20 steps on three surrogate pre-trained models from [github](https://github.com/huyvnphan/PyTorch\_CIFAR10): IncV3 (InceptionV3), VGG19, and DN201 (DenseNet201)**.
>
> The following test robustness shows that AT boosts the robustness against transferable black-box attacks, and $A^2$ can further improve the adversarial robustness.
>
> || IncV3	| VGG19 |DN201| PGD20|
> |---|----|---|---|---|
> |ResNet-18|16.12|7.37|5.35| 0.02|
> |ResNet-18-AT	|61.98	|60.81	|59.63	|52.79|
> |ResNet-18-AT-$A^2$	|**62.79**| **61.85**| **60.28** | **52.96** |
>
>
> Moreover, AutoAttack includes a query-efficient black-box attacker (i.e., Square Attack).
>
>
>
> ---
>
> > Q3&Limitation1: Is there a combination of attack types or step sizes that are clearly selected a vast majority of the time? If so, does provide a considerable improvement over just using this combination of attack parameters? Or are there classes or datasets that are more vulnerable to certain attack combinations?
>
>
> Thanks a lot for the comment.
>
> We analyze the selected attacks from the perspective of blocks with different steps and datasets.
>
> The first and final perturbation blocks of 10-step $A^2$ in CIFAR-10 are chosen for analysis.
> Figures in Appendix B.5 show the distribution of selected attacks of different perturbation blocks.
>
> - **Perturbation Block 1:** $A^2$ tends to choose FGM, FGSM, and partially random methods as initialization in the first step.
> The momentum-based attack methods are quickly discarded as the gradient of the previous step is absent.
> FGSM is chosen more frequently due to its stronger attack on both foreground and background.
> - **Perturbation Block 10:** The optimization of the victim model leads to changes in the distribution of selected attacks in the last block.
> In the early stage of training, the victim model is vulnerable.
> $A^2$ retains the diversity and plays the role of friendly attackers like FAT[5].
> At the end of the training, $A^2$ prefers the momentum-based attacks (i.e., FGSMM and FGMM).
>
> From the perspective of datasets, SVHN and CIFAR-10 prefer different attack methods.
> SVHN discards FGSMM, which is most frequently used in CIFAR-10, and pays more attention to FGMM.
>
> In summary, $A^2$'s preference for selecting attacks in blocks varies according to the block step, dataset, and victim model.
>
>
> ---
>
> > Q1& Limitation2: Are the results from tables 2, 3, and 4 also run 5 times and averaged as well?
>
> Yes. As details, we run 5 times for Table1&Table4. For Table2&Table3, limited by the huge resources that adversarial training consumes, we run the attack to test adversarial robustness 5 times. We have highlighted this in the table caption.
>
> For reproducibility, we provide the source code and scripts with fixed random seeds in SupplementaryMaterial.
>
>
> > Limitation3: Emphasizing or explaining the significance of the relatively small performance improvements afforded by $A^2$.
>
> Thank you for the suggestion.
> In the revised version, we have emphasized the significance of performance improvements.
>
> - **Test Robustness:**
> $A^{2}$ generates stronger perturbations with low extra cost and improves the robustness of various AT methods against different attacks.
> Compared to the recent related works [4][5][6], the 1~2 percent robustness improvement on CW and AutoAttack introduced by $A^2$ is already non-trivial.
> - **Attack Effect:**
> Considering generating perturbations on-the-fly during training, we balance the attack effect with the overhead.
> At 1/5 cost, the 20-step $A^2$ finds better attacks than PGD100.
> The results in Appendix B.3. "Generality of $A^2$ in White-Box Attacks" show that $A^2$ is general to improve the attack effects of PGD and CW.
>
>
>
> [1] Dong, et al. "Boosting adversarial attacks with momentum." CVPR 2018.
>
> [2] Xie, et al. "Improving transferability of adversarial examples with input diversity." CVPR 2019.
>
> [3] Dong, et al. "Evading Defenses to Transferable Adversarial Examples by Translation-Invariant Attacks." CVPR 2020.
>
> [4] Wang, et al. "Improving adversarial robustness requires revisiting misclassified examples." ICLR 2019.
>
> [5] Zhang, et al. "Attacks which do not kill training make adversarial learning stronger." ICML 2020.
>
> [6] Wu, et al. "Adversarial weight perturbation helps robust generalization." NeurIPS 2020.

---

> > ### Comment · Reviewer_SX8F · 2022-08-07
> > **Thank you for your response; score increased!**
> >
> > EOM

---

> > > ### Author Response · Authors · 2022-08-07
> > > **Thank you for the acknowledgment**
> > >
> > > Thanks for your insightful suggestions.
> > > The distribution of selected attacks and other issues help a lot in further improving our paper.

---

### Meta-Review · Area_Chair_aK8n · 2022-08-28

**Recommendation:** Accept
**Confidence:** Certain

**Metareview:**

Based on the idea of AutoML, this paper proposes an attack method that efficiently generates strong adversarial perturbations. The main idea is to use an attention mechanism to score possible attacks in the attacker space, then sample the attack to perform based on the assigned scores. The experimental results show that the proposed method can increase the attack power and improve the adversarial training performance without too much overhead. The reviewers suggest the authors to release code to help others reproduce the results.

**Award:**

No

---

### Decision · Program_Chairs · 2022-09-14

Accept